# Coexistence of Myelin Oligodendrocyte Glycoprotein Immunoglobulin G and Neuronal or Glial Antibodies in the Central Nervous System: A Systematic Review

**DOI:** 10.3390/brainsci12080995

**Published:** 2022-07-27

**Authors:** Cong Zhao, Pei Liu, Daidi Zhao, Jiaqi Ding, Guangyun Zhang, Hongzeng Li, Jun Guo

**Affiliations:** 1Department of Neurology, Air Force Medical Center of PLA, Beijing 100142, China; zhaocongfmmu@126.com (C.Z.); zhgyunfmmu@163.com (G.Z.); 2Department of Neurology, Tangdu Hospital, Air Force Medical University, Xi’an 710038, China; zdd198503@163.com (D.Z.); jangelding0212@163.com (J.D.); llihongzeng@163.com (H.L.); 3Department of Neurology, Xi’an City First Hospital, Xi’an 710002, China; liupei32@outlook.com

**Keywords:** coexistence, antibodies, myelin oligodendrocyte glycoprotein, N-Methyl-D-Aspartate Receptor, systematic review

## Abstract

Background: Myelin oligodendrocyte glycoprotein immunoglobulin G (MOG-IgG) has been considered a diagnostic marker for patients with demyelinating disease, termed “MOG-IgG associated disorder” (MOGAD). Recently, the coexistence of MOG-IgG and other neuronal or glial antibodies has attracted extensive attention from clinicians. In this article, we systematically review the characteristics of MOG-IgG-related antibody coexistence syndrome. Methods: Two authors independently searched PubMed for relevant studies published before October 2021. We also manually searched the references of each related article. The appropriateness of the included studies was assessed by reading the titles, abstracts, and full texts if necessary. Results: Thirty-five relevant publications that met our inclusion criteria were finally included, of which fourteen were retrospective studies and twenty-one were case reports. A total of 113 patients were reported to show the coexistence of MOG-IgG and neuronal or glial antibodies. Additionally, 68.14% of patients were double positive for MOG-IgG and N-Methyl-D-Aspartate Receptor-IgG (NMDAR-IgG), followed by 23.01% of patients who were double positive for MOG-IgG and aquaporin4-IgG (AQP4-IgG). Encephalitis was the predominant phenotype when MOG-IgG coexisted with NMDAR-IgG, probably accompanied by imaging features of demyelination. Patients with dual positivity for MOG-IgG and AQP4-IgG experienced more severe disease and more frequent relapses. The coexistence of MOG-IgG and antibodies other than NMDAR-IgG and AQP4-IgG was extremely rare, and the clinical presentations were diverse and atypical. Except for patients who were double positive for MOG-IgG and AQP4-IgG, most patients with multiple antibodies had a good prognosis. Conclusions: MOG-IgG may coexist with neuronal or glial antibodies. Expanded screening for neuronal or glial antibodies should be performed in patients with atypical clinical and radiological features.

## 1. Introduction

Myelin oligodendrocyte glycoprotein (MOG), uniquely expressed on oligodendrocytes, is located on the outermost layer of the myelin sheath and might act as an adhesion molecule, a regulator of cell skeletal stability, or an activator of complement [1,2]. It has been regarded as an encephalitogenic protein because it can initiate demyelination in numerous animal models [3,4]. Thanks to the refinement of the new-generation cell-based assays (CBAs), autoantibodies against full-length human MOG protein (MOG-IgG) have been detected in some patients with inflammatory demyelinating diseases (IDDs) of the central nervous system, such as multiple sclerosis (MS) [5], aqueporin4 (AQP4)-IgG-negative neuromyelitis optica (NMO) [6,7], and acute disseminated encephalomyelitis (ADEM) [8]. Recently, accumulating evidence has suggested that MOG-IgG-positive patients have clinical characteristics distinct from other IDDs, which support MOG-IgG-associated disorder (MOGAD) as a novel independent disease entity [8,9].

Typical clinical phenotypes of MOGAD include optic neuritis (ON), ADEM, transverse myelitis (TM), and brainstem encephalitis [9,10]. In recent years, the spectrum of MOGAD has been expanded due to the detection of MOG-IgG coexisting with other neuronal or glial antibodies, especially in patients with atypical clinical symptoms and/or neuroradiological features [11]. Our group recently reported two patients with atypical MOGAD in whom MOG-IgG coexisted with glial fibrillary acidic protein (GFAP)-IgG [12] and contact protein-associated 2 (CASPR2)-IgG [13], respectively. An increasing number of studies have also demonstrated the coexistence of MOG-IgG with other antibodies, such as N-Methyl-D-Aspartate Receptor (NMDAR)-IgG [14] and AQP4-IgG [15,16], which has drawn extensive attention and generated discussion. However, due to its rarity, previous studies on antibody coexistence syndrome were either case reports or small sample studies, making it difficult to reach consistent conclusions. Therefore, this systematic review aims to summarize the existing literature to analyze the characteristics of MOG-IgG-related antibody coexistence syndrome and discuss the possible mechanism of poly-immunoreactivity in MOG-IgG-positive patients.

## 2. Methods

### 2.1. Search Strategy and Study Selection

Our study was conducted according to the Preferred Reporting Items for Systematic Reviews and Meta-Analysis (PRISMA) guideline [17]. Two authors (Cong Zhao and Pei Liu) independently searched PubMed using a combination of medical subject headings and search words as follows: (“Myelin-Oligodendrocyte Glycoprotein” OR “MOG”) AND (“coexist*” OR “dual positive” OR “double positive” OR “overlap”). Details of the search strategy are provided in the Appendix A. The search was limited to articles published before October 2021. Additionally, we manually searched the references of the included studies.

The appropriateness of studies for their inclusion was assessed by two authors (Cong Zhao and Daidi Zhao) by reading the titles, abstracts, and, if necessary, the full texts. The inclusion criteria were as follows: (1) We included patients in whom MOG-IgG coexisted with other autoimmune antibodies targeting the central nervous system (CNS), detected in either the serum or CSF. Antibodies could appear simultaneously or successively. (2) The included studies were retrospective studies or case reports published in English. Reviews and studies reporting animal and molecular experiments were excluded.

### 2.2. Data Extraction

For each study, the author’s name, publication date, study design, and country were extracted. The following characteristics of patients were recorded when available: the number of patients, age, gender, follow-up duration, the presence of CNS autoantibody spectrums, antibody titers, clinical manifestations, neuroimaging characteristics, treatment regimens, and long-term outcome. Data were collected independently by two authors (Cong Zhao and Pei Liu), and any disagreements were discussed with a third author (Jiaqi Ding) until consensuses were reached.

### 2.3. Statistical Analysis

The incidence of clinical symptoms and lesion distribution among MOG-IgG and NMDAR-IgG dual-positive episodes, MOG-IgG single-positive episodes, and NMDAR-IgG single-positive episodes were compared by Fisher’s exact test using SPSS 27.0 (IBM Corporation, Armonk, NY, USA). Values of *p* < 0.05 were considered statistically significant.

## 3. Results

### 3.1. Study Characteristics and the Spectrum of Coexistence Syndromes

The process of study selection is shown in Figure 1. Overall, a total of 104 records were obtained after database searching. Fifty-three relevant studies were identified after title and abstract screening. After reading the full texts and reviewing the references of the retrieved articles, 35 studies were finally included in the qualitative synthesis, of which 14 were retrospective studies [11,15,16,18,19,20,21,22,23,24,25,26,27,28] and 21 were case reports [12,13,29,30,31,32,33,34,35,36,37,38,39,40,41,42,43,44,45,46,47]. A total of 113 patients (46 males and 67 females) were reported to show the coexistence of MOG-IgG and neuronal or glial antibodies in these 35 studies. The age of onset ranged from 2 to 66 years, with a median of 24 years.

The spectrum of the coexistence of MOG-IgG and neuronal or glial antibodies is shown in Figure 2. NMDAR-IgG was the most frequently coexisting antibody with MOG-IgG. Eighty patients (70.80%) showed the coexistence of MOG-IgG and NMDAR-IgG, of which seventy-seven cases (68.14%) were double positive for MOG/NMDAR-IgG, and the remaining three cases were triple positive for MOG/NMDAR/CASPR2-IgG, MOG/NMDAR/AQP4-IgG, and MOG/NMDAR/GFAP-IgG, respectively. Twenty-six (23.01%) patients were double positive for MOG-IgG and AQP4-IgG, which was the second most common coexistence syndrome. The remaining antibody coexistence syndromes included MOG-IgG coexisting with GFAP-IgG (two patients, 1.77%), MOG-IgG with CASPR2-IgG (two patients, 1.77%), MOG-IgG with leucine-rich glioma inactivated 1 (LGI1)-IgG (one patient, 0.88%), MOG-IgG with gamma-aminobutyric acid-A receptors (GABAA-R)-IgG (one patient, 0.88%), and MOG-IgG with GFAP-IgG and inositol 1,4,5-triphosphate receptor type 1 (ITPR-1)-IgG (one patient, 0.88%). Other antibody coexistence syndromes included MOG-IgG coexisting with GFAP-IgG (two patients, 1.77%), MOG-IgG coexisting with CASPR2-IgG (two patients, 1.77%), MOG-IgG coexisting with leucine-rich glioma inactivation 1 (LGI1)-IgG (one patient, 0.88%), MOG-IgG coexisting with γ-Aminobutyric acid-a receptor (GABAA-R)-IgG (one patient, 0.88%), MOG-IgG coexisting with GFAP-IgG (one patient, 0.88%), and MOG-IgG coexisting with inositol 1,4,5-triphosphate receptor type 1 (ITPR-1)-IgG (one patient, 0.88%).

### 3.2. Coexistence of MOG-IgG and NMDAR-IgG

The combined demographic and clinical characteristics of patients who were double positive for MOG-IgG and NMDAR-IgG are listed in Table 1. The median age at onset was 21 years (age range, 2~63 years), and 40.28% of patients were female (with sex not specified in five patients). The median titer of MOG-IgG in serum was 1:100 (range, 1:10~1:16,384), and the median titer of NMDAR-IgG in CSF was 1:32 (range, 1:1~1:320). The median follow-up duration after diagnosis was 15 months (range, 2~144 months). Most of the patients had a favorable outcome after immunotherapy. The Modified Rankin Scale (mRS) score was documented in 28 patients, 26 (92.86%) of whom scored less than or equal to 2 at the last follow-up.

In the included literature, a total of 70 patients had detailed clinical data available for analysis. Among them, 45 patients (64.29%) experienced relapsing-remitting courses, and 25 patients (35.71%) had a monophasic course before the last follow-up. Additionally, 52 patients (74.29%) were positive for MOG-IgG and NMDAR-IgG simultaneously in a single episode, 28 of whom showed double positivity in the first episode. Of the total 104 episodes recorded in these 70 patients, MOG-IgG and NMDAR-IgG appeared together in 47 episodes, MOG-IgG alone was detected in 26 episodes, NMDAR-IgG alone was identified in 21 episodes, and antibody status in the remaining 10 episodes was unknown.

The clinical symptoms and imaging features of each episode were analyzed as follows. As shown in Figure 3, when MOG-IgG or NMDAR-IgG was present alone, the clinical characteristics were dominated by demyelination or encephalitis, respectively. Among the clinical episodes in which MOG-IgG appeared simultaneously with NMDAR-IgG, the clinical manifestations remained similar to the symptom spectrums associated with anti-NMDAR encephalitis, i.e., psychiatric symptoms (33/47, 70.21%), seizures (22/47, 46.81%), speech disorders (20/47, 42.55%), consciousness disorders (12/47, 25.53%), autonomic dysfunction (11/47, 23.40%), and movement disorders (11/47, 23.40%). In addition, a considerable number of episodes also presented with demyelinating syndromes, such as optic neuritis (9/47, 19.15%), brainstem syndromes (9/47, 19.15%), and myelitis (4/47, 8.51%). Statistical analysis revealed significant differences in the incidence of psychiatric symptoms (*p* < 0.001), seizures (*p* = 0.007), headaches (*p* = 0.019), autonomic dysfunction (*p* = 0.015), speech disorders (*p* = 0.016), and brainstem syndromes (*p* = 0.018) among the three types of clinical episodes. The heatmap in Figure 3 intuitively illustrates that the clinical spectrum of dual-positive episodes, although approximating the combination of symptoms when both antibodies occurred alone, was still dominated by symptoms associated with anti-NMDAR encephalitis.

The lesion distribution for each episode is demonstrated in Figure 4. In episodes with MOG-IgG single positivity, lesions were mostly located in the infratentorial structure and deep gray matter, such as the pons (9/26, 34.62%), midbrain (5/26, 19.23%), thalamus (6/26, 23.08%), and basal ganglia (5/26, 19.23%), while the frontal (6/21, 28.57%) and temporal cortexes (7/21, 33.33%) were mostly affected in attacks with NMDAR-IgG single positivity. Imaging characteristics were more complex when the above two antibodies existed together. The lesions were more widely distributed and predominant in subcortical white matter (16/47, 34.04%), basal ganglia (15/47, 31.91%), frontal cortex (13/47, 27.66%), and temporal cortex (13/47, 27.66%), as well as the involvement of infratentorial structures such as the midbrain (13/47, 27.66%), pons (11/47, 23.40%), and spinal cord (12/47, 25.53%). The incidence of temporal cortex (*p* = 0.012) and midbrain (*p* = 0.014) lesions was significantly different among the three clinical episode types.

### 3.3. Coexistence of MOG-IgG and AQP4-IgG

Previous studies reported that MOG-IgG rarely coexisted with AQP4-IgG in a single patient [15]. After a thorough literature search, 26 patients were found to be double positive for MOG-IgG and AQP4-IgG. The main characteristics of the included patients are summarized in Table 2. Females accounted for an extremely high proportion of double-positive patients. The median onset age was 35 years (range, 15–66 years). The disease course was reported in 14 patients. Thirteen (92.9%) patients experienced multiphasic disease. The median disease duration was 4 years (range, 2–11 years), and the median number of attacks was 6 (range, 1–11). Residual disability was severe, with a median Expanded Disability Status Scale (EDSS) score of 8 (range, 5–9). Antibody titers were reported in 10 patients, with higher titers for AQP4-IgG (median, 1:10,000) and lower titers for MOG-IgG (median, 1:40) [15].

Yan et al. compared the characteristics of double-positive patients with those of single-positive patients [16]. Patients who were double positive for MOG-IgG and AQP4-IgG had more severe disease than single-positive patients. The nadir EDSS score of double-positive patients was significantly higher and decreased less after treatment than that of single-positive patients. Their annual relapse rate (ARR) was also higher. All patients developed recurrent optic neuritis and longitudinally extensive transverse myelitis. Brain involvement was observed in all 10 patients, including 7 patients with MS-like lesions and 3 patients with ADEM-like lesions. The conus was more likely to be affected in double-positive patients. Severe atrophy of the optic nerve and a reduction in retinal nerve fiber layer thickness were also observed in double-positive patients.

### 3.4. Coexistence of MOG-IgG and Other Neuronal or Glial Antibodies

In addition to the two types of antibody coexistence syndromes mentioned above, only 10 cases of MOG-IgG coexisting with other neuronal or glial antibodies were reported. The clinical and radiological data of each patient are listed in Table 3. Encephalitis seemed to be the main clinical syndrome when MOG-IgG coexisted with neuronal antibodies, as shown in cases 1 to 4. However, when MOG-IgG coexisted with glial antibodies (cases 6 to 10), demyelination was dominant, such as optic neuritis, myelitis, and brainstem encephalitis.

## 4. Discussion

Our study provided a comprehensive description of patients who had coexisting MOG-IgG and neuronal or glial antibodies. After analyzing the included literature, we summarized the key findings as follows: (1) Currently, NMDAR-IgG is the most common antibody that coexists with MOG-IgG, followed by AQP4-IgG. (2) The clinical phenotype of patients with MOG-IgG and NMDAR-IgG double positivity was dominated by anti-NMDAR encephalitis, and a proportion of patients may suffer complications with demyelination events. Most patients had a good prognosis after immunotherapy. (3) Patients with coexisting AQP4-IgG and MOG-IgG had more severe symptoms, more frequent recurrence, and a higher degree of disability. (4) The coexistence of MOG-IgG with other neuronal or glial antibodies was extremely rare, which requires further study.

The incidence of coexistence syndromes varied among studies. Titulaer et al. first reported that 12 patients (1.7%) were positive for MOG-IgG in a cohort of 691 patients with anti-NMDAR encephalitis [26]. Subsequent retrospective studies reported that MOG-IgG could be detected in 2.0% to 14.2% of patients with anti-NMDAR encephalitis [18,19,23], and NMDAR-IgG was detected in 2.9% to 11.9% patients with MOGAD [11,20,27]. Pooled data from a recent meta-analysis demonstrated that approximately 9% of MOG-IgG-positive patients had coexisting NMDAR-IgG, and 7% of patients with anti-NMDAR encephalitis were positive for MOG-IgG [14]. Kunchok et al. reported that out of a cohort of 1250 patients positive for MOG-IgG or AQP4-IgG, only 10 patients (0.8%) presented double positivity [15]. Thus, it seems that NMDAR-IgG is more likely to coexist with MOG-IgG than AQP4-IgG. Moreover, MOG-IgG rarely coexisted with other antibodies such as GFAP-IgG, LGI1-IgG, CASPR2-IgG, and GABAA-R-IgG [11], suggesting an exclusive relevance with NMDAR-IgG.

The clinical manifestations of antibody coexistence syndrome are atypical and complex, which may lead to misdiagnosis or underdiagnosis. Previous retrospective studies showed that the clinical phenotype of patients with double positivity for MOG-IgG and NMDAR-IgG was more prone to encephalitis when compared to MOGAD patients without NMDAR-IgG [11]. Since some patients had multiphasic courses, we analyzed the characteristics of each episode according to the antibody profile. The symptoms of most double-positive episodes were highly similar to those of anti-NMDAR encephalitis. Among these symptoms, psychiatric disorders, seizures, and speech disorders were the most common. Unlike typical anti-NMDAR encephalitis, double-positive patients/episodes had milder disease and were less likely to progress to status epilepticus or unconsciousness [20,26]. Furthermore, demyelinating events, such as optic neuritis, myelitis, and brainstem encephalitis, may also present in a proportion of double-positive episodes. Consistent with our results, Titulaer et al. found that half of the cases included in their cohort developed at least one episode of demyelination, which suggested the pathogenicity of MOG-IgG [26]. Imaging features were more complicated than clinical symptoms. In addition to the frontotemporal cortex, which was commonly affected in anti-NMDAR encephalitis, a broad range of structures were involved in double-positive episodes, including the infratentorial regions, subcortex, and basal ganglia [19,20,21,29,35,36,41]. Recent studies showed that lesions in MOGAD were typically located infratentorially [20,48]. Our results also suggested that the frequency of midbrain lesions was higher in MOG-IgG-positive episodes than in NMDAR-IgG-positive episodes. Therefore, patients that exhibit psychiatric disorders or seizures as well as demyelination and atypical brain lesions, especially infratentorial lesions, should be screened for MOG-IgG and NMDAR-IgG together.

The underlying mechanisms of poly-immunoreactivity have not been clearly elucidated. The coexistence of multiple antibodies may be partially explained by the concept of epitope spreading; that is, persistent recognition and reaction to one epitope of an antigen may result in the spread of the immune response to other epitopes (intramolecular spreading) or to other antigens (intermolecular spreading) [49,50]. MOG-IgG may present consecutively with other neuronal or glial antibodies, suggesting the involvement of intermolecular epitope spreading from MOG to other proteins, or vice versa. Another hypothesis is that direct viral infection of the brain leads to the breakdown of the blood–brain barrier and subsequent infiltration of immune cells into CNS, where they recognize and attack multiple antigens and cause antigen leakage into the peripheral circulation. Malignant tumors are usually associated with autoimmune encephalitis and are considered to be immune triggers for autoimmune responses. Nearly half of patients with anti-NMDAR encephalitis have ovarian teratomas [51,52]. However, few patients who were double positive for MOG-IgG and NMDAR-IgG had teratomas, suggesting that tumors may not be the cause of antibody coexistence syndrome [20,26].

The diagnosis of patients with double or multiple antibodies is usually challenging. In our opinion, diagnosis should rely on a combination of clinical symptoms, radiological characteristics, and antibody profiles. Identifying which antibodies are culprits and which are bystanders is important for diagnosis and immunotherapy. The clinical manifestations of some patients with antibody coexistence syndrome are only related to one antibody. For example, Sarigecili reported that a boy who was double positive for NMDAR-IgG and MOG-IgG experienced alterations in behavior, gait, and speech but did not have any demyelinating symptoms or imaging changes [43]. In this case, MOG-IgG might be a bystander, and MOGAD should not be diagnosed. Moreover, patients with MOG-IgG and AQP4-IgG coexistence were similar to those with NMOSD in terms of both the disease recurrence rate and disability [16]. The titer of AQP4-IgG in patients with MOG-IgG and AQP4-IgG coexistence was much higher than that of MOG-IgG [15]. Therefore, AQP4-IgG should be considered dominant in these patients. We believe that more antibody coexistence syndromes will be detected with the popularity of the CBA method. However, antibody status should never be the only criterion for diagnosis. The clinical phenotype always takes priority.

There are several limitations of this systematic review. First of all, the sample size was too small to show the full picture of the disease. Secondly, the included studies were case reports and retrospective studies from different countries and institutes. Patients were evaluated differently and reported heterogeneously in each study. This heterogeneity made it difficult to draw comprehensive conclusions. Finally, there were too few reports of MOG-IgG coexisting with neuronal or glial antibodies other than NMDAR-IgG and AQP4-IgG.

## 5. Conclusions

MOG-IgG may coexist with neuronal or glial antibodies, including NMDAR-IgG, AQP4-IgG, GFAP-IgG, CASPR2-IgG, LGI1-IgG, and GABAAR-IgG. MOG-IgG and NMDAR-IgG coexistence is the most common antibody coexistence syndrome, which may manifest as encephalitis and demyelination. Most patients with multiple antibodies have a good prognosis, except for patients who are double positive for MOG-IgG and AQP4-IgG. Broad antibody screening should be carried out in patients with atypical clinical and radiological features. Given the limitations of the present study, further multi-center prospective research and animal studies are required to comprehensively understand this phenomenon.

## Figures and Tables

**Figure 1 brainsci-12-00995-f001:**
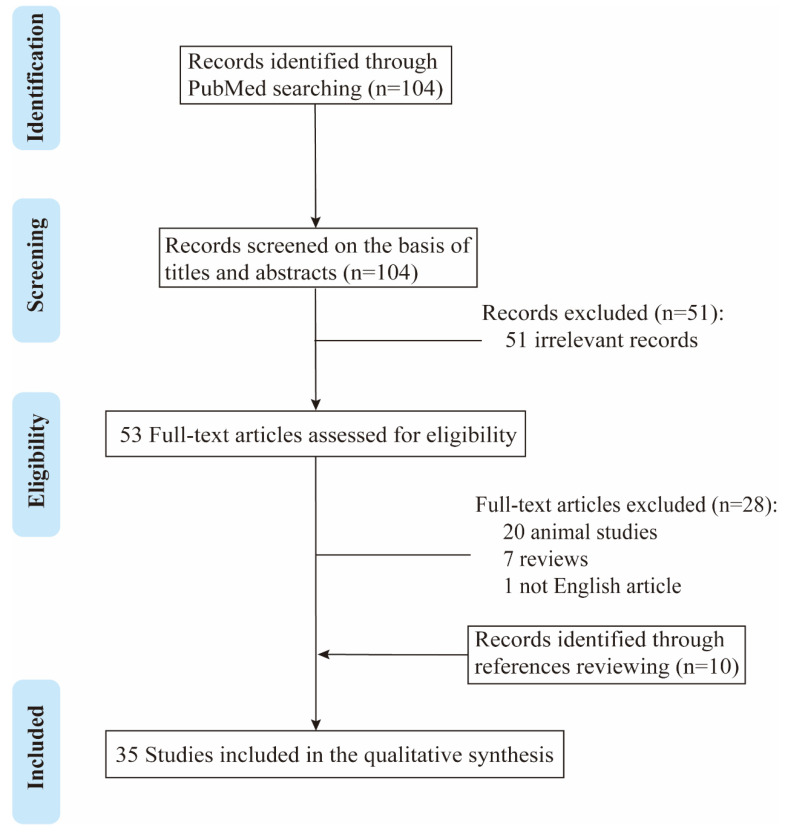
Flow chart of study selection algorithm according to PRISMA guidelines.

**Figure 2 brainsci-12-00995-f002:**
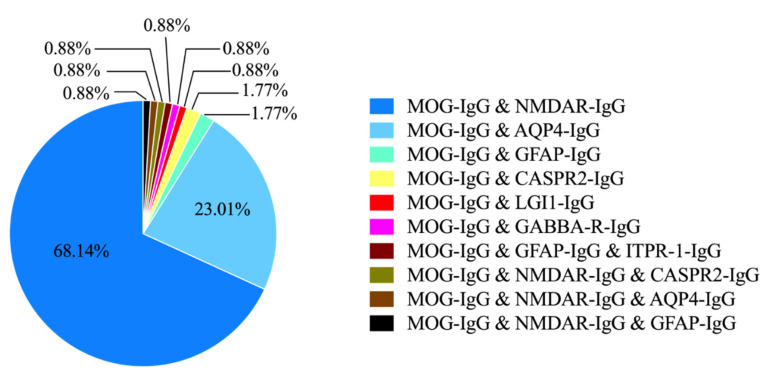
Percentages of antibody coexistence syndromes.

**Figure 3 brainsci-12-00995-f003:**
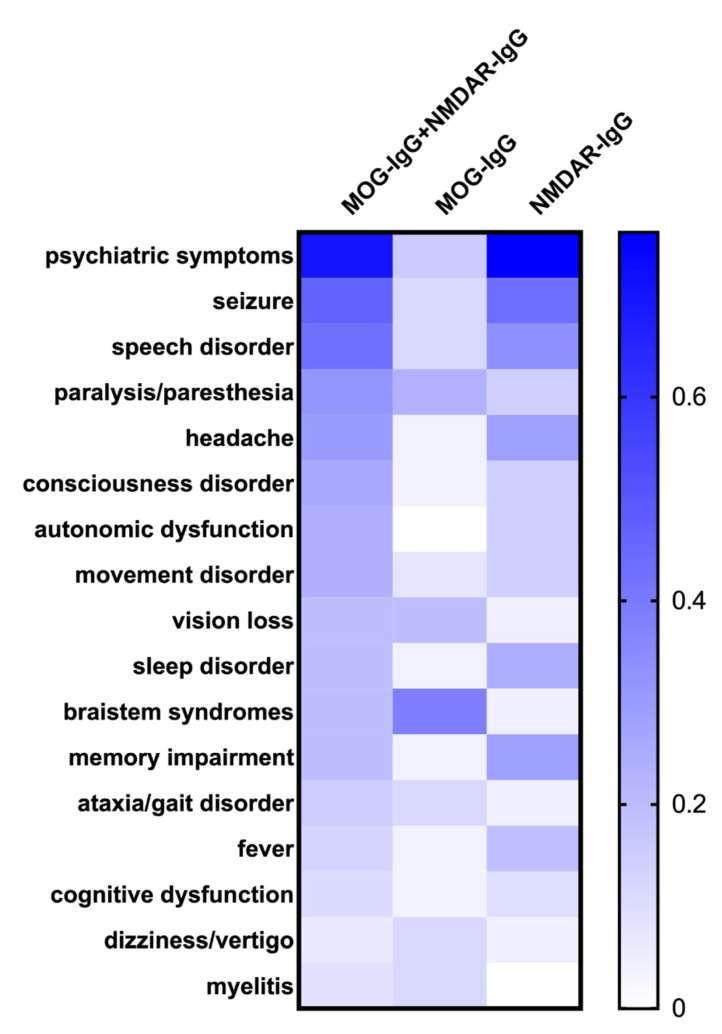
Heatmap of clinical symptoms of MOG-IgG and NMDAR-IgG dual-positive episodes, MOG-IgG single-positive episodes, and NMDAR-IgG single-positive episodes.

**Figure 4 brainsci-12-00995-f004:**
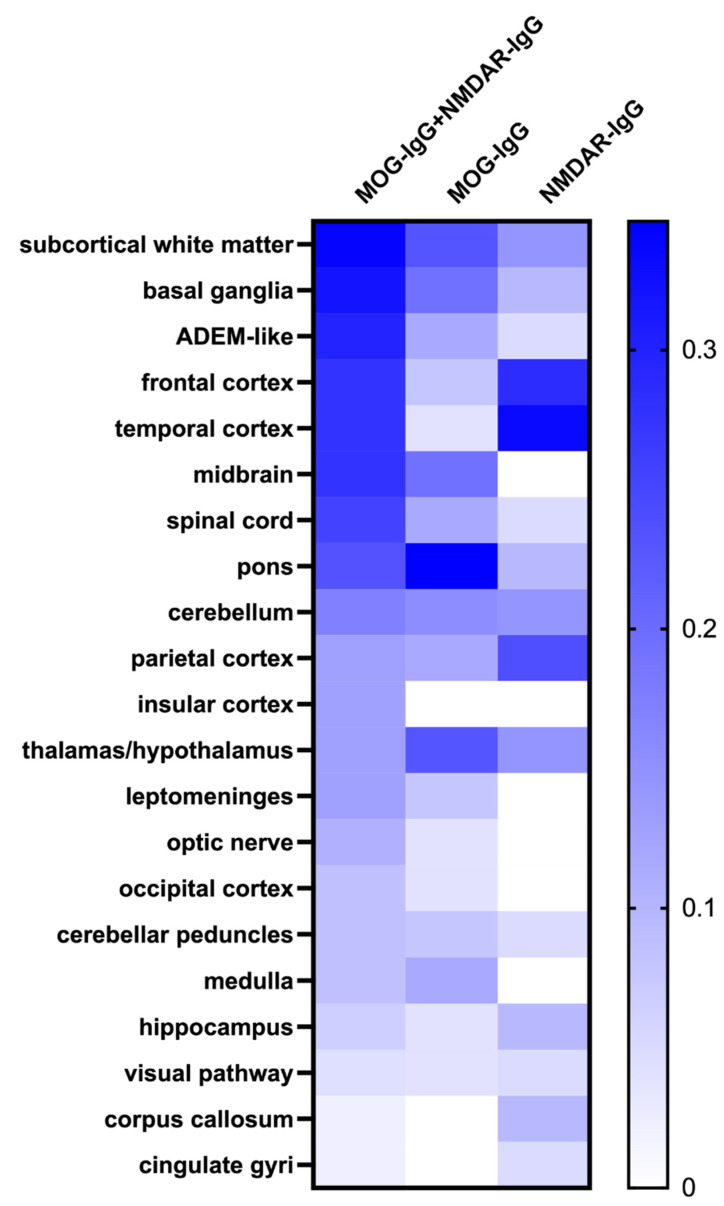
Heatmap of lesion distribution of MOG-IgG and NMDAR-IgG dual-positive episodes, MOG-IgG single-positive episodes, and NMDAR-IgG single-positive episodes.

**Table 1 brainsci-12-00995-t001:** Features of patients double positive for MOG-IgG and NMDAR-IgG.

Features	
Age at onset, median (range), y	21 (2~63)
Female, n (%) ^a^	29 (40.28)
Titer of serum MOG-IgG ^b^	1:100 (1:10~1:16,384)
Titer of CSF NMDAR-IgG ^c^	1:32 (1:1~1:320)
Follow-up duration, median (range), mo^d^	15 (2~144)
mRS score ≤ 2 at last follow-up, *n* (%) ^e^	26 (92.86)

mRS: Modified Rankin Scale. ^a^: available for 72 patients; ^b^: available for 28 patients; ^c^: available for 34 patients; ^d^: available for 25 patients; ^e^: available for 28 patients.

**Table 2 brainsci-12-00995-t002:** Features of patients double positive for MOG-IgG and AQP4-IgG.

Features	
Age at onset, median (range), y ^a^	35 (15–66)
Female, *n* (%) ^b^	23 (95.8)
Disease duration, median (range), y ^c^	4 (2–11)
EDSS score at last follow-up ^d^	8 (5–9)
Attack number ^e^	6 (1–10)
OCB, *n* (%) ^f^	1 (8.3)

EDSS, Expanded Disability Status Scale; OCB, oligoclonal bands. ^a^: Available for 14 patients; ^b^: available for 24 patients; ^c^: available for 11 patients; ^d^: available for 11 patients; ^e^: available for 14 patients; ^f^: available for 12 patients.

**Table 3 brainsci-12-00995-t003:** Clinical and radiological data of patients with MOG-IgG coexisting with other neuronal or glial antibodies.

No./Age/Gender	Coexisting Antibodies	Clinical Manifestation	Imaging Features
1/48/F (13)	CASPR2-IgG	Decreased vision, dizziness, speech disorder, gait instability, urinary incontinence, psychiatric symptoms	Hyperintensities in cortex, cerebral peduncle, brainstem, thalamus, corpus callosum, cervical and thoracic spinal cord
2/10/M (11)	CASPR2-IgG	Ascending paralysis, intractable seizures	ADEM-like lesions involving bilateral hemisphere, brainstem, and cerebral peduncle
3/30/M (37)	NMDAR-IgG and CASPR2-IgG	Headache, psychological and behavioral abnormalities, memory loss, cerebellar dysarthria, spastic ataxia	Hyperintensities in bilateral cingulate gyri, hippocampus, pulvinar; patchy perivascular and subpial enhancement over pons, cerebellar peduncle, cerebellar folia, midbrain, and cingulate gyri
4/59/M (11)	GABAA-R-IgG	Focal seizures, encephalopathy	Hyperintensities in bilateral temporal lobe
5/55/F (11)	LGI1-IgG	NA	NA
6/20/M (12)	GFAP-IgG	Decreased vision, diplopia, nystagmus, dizziness, hemiplegia, Romberg’s sign	Swelling of bilateral ON; hyperintense patchy lesions in cerebellum, brachium pontis, and temporal lobe
7/23/F (42)	GFAP-IgG	Fever, headache, vomiting, convulsion, Kernig sign	Diffuse leptomeningeal enhancement; asymmetric hyperintense signal in cerebellum and corona radiata, radial enhancement patterns extending outward from the ventricles
8/27/F (11)	AQP4-IgG and NMDAR-IgG	Optic neuritis, cervical LETM	Lesions in temporal lobe, thalamus, optic tract and chiasm, spinal cord
9/33/M (11)	GFAP-IgG and NMDAR-IgG	First attack: multifocal meningoencephalitis; second attack: cervical LETM	First attack: T2 hyperintensity and leptomeningeal enhancement along left temporal, frontal, and parietal cortexes; second attack: hyperintensities in basal ganglia, cerebellar peduncle, and spinal cord
10/44/F (40)	GFAP-IgG and ITPR-1-IgG	Fever, nausea, vomiting, paraplegia, ataxia, nystagmus, urinary retention, respiratory paralysis	Signs of meningitis, cortical and subcortical lesions within parietooccipital cortex with diffuse restriction, edema of medulla oblongata

LETM, longitudinally extensive transverse myelitis; ADEM, acute disseminated encephalomyelitis; ON, optic neuritis; NA, not available.

## Data Availability

Not applicable.

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
