# Peer review of "Coexistence of Myelin Oligodendrocyte Glycoprotein Immunoglobulin G and Neuronal or Glial Antibodies in the Central Nervous System: A Systematic Review"

_brainsci, 2022, doi:10.3390/brainsci12080995_

Round 1

Reviewer 1 Report

English language and style are  spell check required for a better understanding of the text

Author Response

Dear Reviewer,

Thanks for your suggestions.

We have carefully read the full text, polished the language, and corrected some grammatical errors and ambiguous expressions. All the revisions made in the manuscript were marked up using "Track Changes" function. We also added abbreviations when some phrases first appeared.

Thank you for reviewing our manuscript.

Reviewer 2 Report

Dear authors, 

Congratulations for this interesting systematic review, settled on PRISMA guidelines. I found it well done and supported by statistical analysis.

Author Response

Dear Reviewer,

Thank you very much for your affirmation.

We have read through the full text and polished the language, and some grammatical errors were corrected.

Thank you for reviewing our manuscript!

Reviewer 3 Report

Overall, this manuscript is well written, has an important clinical message, and should be of great interest to the readers.

Author Response

(The authors gave the same response as above.)
